# The Effects of Transparency and Reliability of In-Vehicle Intelligent Agents on Driver Perception, Takeover Performance, Workload and Situation Awareness in Conditionally Automated Vehicles

Jing Zang and Myounghoon Jeon *

Department of Industrial and Systems Engineering, Virginia Tech, Blacksburg, VA 24061, USA
* Correspondence: myounghoonjeon@vt.edu

**Abstract:** In the context of automated vehicles, transparency of in-vehicle intelligent agents (IVIAs) is an important contributor to driver perception, situation awareness (SA), and driving performance. However, the effects of agent transparency on driver performance when the agent is unreliable have not been fully examined yet. This paper examined how transparency and reliability of the IVIAs affect drivers' perception of the agent, takeover performance, workload and SA. A 2 × 2 mixed factorial design was used in this study, with transparency (*Push*: proactive vs. *Pull*: on-demand) as a within-subjects variable and reliability (high vs. low) as a between-subjects variable. In a driving simulator, 27 young drivers drove with two types of in-vehicle agents during the conditionally automated driving. Results suggest that transparency influenced participants' perception on the agent and perceived workload. High reliability agent was associated with higher situation awareness and less effort, compared to low reliability agent. There was an interaction effect between transparency and reliability on takeover performance. These findings could have important implications for the continued design and development of IVIAs of the automated vehicle system.

**Keywords:** automated vehicle; explainable AI; situation awareness (SA); transparency; reliability; trust

## 1. Introduction

As the automation of vehicles increases, it becomes more difficult for drivers to maintain their situation awareness and understand the actions of the system. If a vehicle system works as a "black box", its decision-making process can remain opaque to drivers [1]. When experiencing automated vehicles above Level 3 [2], drivers are able to decouple from the operational and tactical levels of control [3]. Two of the most essential tasks for drivers are: (1) maintaining situation awareness to ensure it to perform as expected and (2) getting for resuming control (i.e., takeover) when the automation deviates from their expectation [2]. A lack of understanding of the reasoning process behind the output of an intelligent system would be a risk for drivers in a dynamic environment in an automated vehicle. For example, if the system only says, "an obstacle ahead", a driver might consider the system to do some action, even though the system wanted the driver to take an action. The lack of a clear explanation of its intent may cause the driver to miss an important maneuver. Thus, the challenge of designing automation to better understand how the automation interacts with human operators, and to determine how the operational information is delivered to the operator has recently come into view.

Researchers have suggested that introducing an *explanation* of algorithmic decision-making can mitigate these adverse effects of black-box-like systems, and foster the driver's understanding and trust, leading to an increase in road safety [1,4]. This explainable AI system then became a concept called XAI, which is used to refer to a system that makes its behavior more intelligible to human operators by providing explanations [5]. As an

anthropomorphized intelligent system that can interact with drivers using natural human language, an in-vehicle intelligent agent (IVIA) is considered a great information delivery medium to support transparent human–agent interactions with automated systems [6]. Chen and colleagues [7,8] proposed the situation awareness-based agent transparency theory (SAT) to provide a framework of what and how information should be delivered to improve the operator's situation awareness (SA), to build a more "transparent" system. Based on the theory of SA [9], the situation awareness-based agent transparency (SAT) model was developed to provide a framework of what information should be delivered to the driver and how the information should be structured to support situation awareness [10]. The SAT model is comprised of three levels of information [8]: Level 1 is the basic information about the agent's goals and proposed actions; Level 2 provides information about the agent's reasoning process behind these actions or plans; and Level 3 is the information regarding the agent's projected outcomes of its current plan (e.g., predicted consequences or likelihood of success/failure).

Given the importance of the topic and research gap in the literature, this study adopted the SAT model to develop transparency levels and test its relationship with the agent's reliability on driver's perception, trust, takeover performance, workload, and situation awareness. The present study is expected to contribute to theory, research design, and practical implementation of in-vehicle agents. First, the results from our study provide profound insights into expanding the theory of explainable AI (artificial intelligence) in dynamic situations such as driving. Second, through the novel factorial design, our study proposed a clear experimental paradigm to examine both transparency and reliability in one study. Finally, the findings of the current study can be used to inspire IVIA design to promote safety as well as user experience in automated vehicles.

The rest of the present paper is organized as follows. Section 2 provides the background for this work. Section 3 develops the research questions. Section 4 describes the method. The results are presented in Section 5 and discussed in Section 6. Section 7 depicts the limitations and future work.

## 2. Related Work

### 2.1. Transparency Design of Interaction

There are many different definitions of what transparency is and how it should be implemented. Transparency can be defined as the amount of information shared with the operator about the function of the system [11]. According to Chen and colleagues [8], transparency enables a user's comprehension of the intention, performance, future actions, and reasoning of automated processes. Such in-depth SAT information can be used to help users determine not only whether an agent made an error, but also why it made an error, which is a crucial consideration to develop appropriate trust and dependence [4,12]. For example, according to Koo and colleagues [3], participants drove more safely when the agent provided explanations for both "how" and "why" the imminent actions were chosen by the automated vehicle. Further, automation shows assistance in relation to reducing attentional demands by providing more information to drivers [13]. Even when implemented appropriately, increased transparency requires additional information in the system's interface, which may increase operator workload [14]. Consequently, the challenge for increasing transparency in a human-automated system is to implement it in a manner that keeps operators in the loop while minimizing additional workload [15].

Furthermore, how the information should be conveyed to the driver is a crucial aspect when designing XAI. Keith et al. [16] argued that a proactive system should be representative of the user and able to initiate behavior without user commands. Proactive interaction can help human operators collect and process information in the environment, thereby, reducing the user's information processing burden [17]. In proactive interaction, human operators can achieve more supervisory control rather than active control [18].

Based on the SAT model, the present study adopted a transparency design of the proactive interaction model: the "Push and Pull" Transparency Model [19,20]. In the Push

condition, all the information, including "what" to do/is happening, "why" the driver should listen, and "how" the system knows what is (going to be) happening/will react was combined in one string and presented to the driver *proactively*; whereas in the Pull condition, although the same type of information would be conveyed to the driver, only the "what" information was initially presented to the driver, and the IVIA could supply the reason behind the current action upon the driver's request [3]. Participants can ask for more information by saying, "*more information*". Because the information is provided in a non-proactive manner in the Pull condition, they would not get full information if they chose not to ask.

## 2.2. Influence of System Reliability on Drivers

Increased system transparency provides the driver with an opportunity to determine the accuracy of the system, which poses a challenge to the reliability of designing the system. Agent reliability is a significant factor that influences the proper use of automation and task performance in multiple ways.

In the present study, IVIA's reliability is defined as the accuracy of information conveyed by IVIA, or its capability. Because the information conveyed by the agent is based on system predictions, there is no way to make sure that it is absolutely accurate (e.g., system malfunction or limitations). On one hand, unreliable automation has been found to have a detrimental impact on a user's task performance, and harm the driver's trust and acceptance of the system [21,22]. On the other hand, highly reliable automation can also have a negative impact on drivers' trust in developing complacency or over-reliance on the system and increasing mental workload [23]. Therefore, controlling the system's reliability within a moderate range could help improve driving performance and avoid over or under trust that leads to misuse or disuse of the vehicle system [24]. Although such unreliability damages drivers' performance to some extent, benefits from an unreliable system were found in a situation in which aircraft predictor information was only partially reliable. Knowing that the predictor was not completely reliable, pilots calibrate their trust and adopt an appropriate allocation of attention between the raw data and the predictor information [25]. In the present study, we examined if and how the reliability level interacts with the system transparency in terms of influencing driver perception and performance in conditionally automated vehicles.

## 2.3. Drivers and In-Vehicle Intelligent Agents

In-vehicle intelligent agents influence drivers' perception of the agent, trust, takeover performance, workload, and situation awareness. Each factor was examined in previous research.

### 2.3.1. Perception of the Agent

Measuring human perception and cognition is an important process when designing human–agent interaction [26]. It is important in how the human perceives the agent, which in turn can affect their acceptance of the automated system. The current study examined how the agent's transparency and reliability affect the driver perception of the humanness and intelligence of the agent [27].

### 2.3.2. Trust

Currently, many studies related to IVIAs for automated driving focus on driver trust. Trust can be defined as the attitude that an agent will help achieve an individual's goals in a situation characterized by uncertainty and vulnerability [4]. An appropriate amount of trust is essential to the effectiveness of the automated system: too much trust will lead to misuse of the system, while lack of trust will result in disuse [24]. Studies have shown that increasing information transparency and system reliability is an effective way to promote trust [7,17,28,29]. The present study examined whether the proactive display of information to support in-depth transparency information would aid in mitigating the impact of the system errors [30].

### 2.3.3. Takeover Performance

The takeover process of the automated vehicle is defined as the transition of control from the automation to a human driver. According to Banks [31], when encountering an emergency takeover scenario, drivers' ability to safely resume control largely depends on the extent to which they remained engaged in monitoring both the automation and external road environment. Because Level 3 automated vehicles require drivers to take over control of the vehicle in certain circumstances, the current study measured takeover performance by obtaining the driving simulator data [32].

### 2.3.4. Workload

Mental workload represents the cognitive resources demanded by a task that is needed to achieve a particular level of performance [33]. In-depth SAT information may increase more complexity of mental processes. A previous study found an increase in mental workload, measured by using the NASA-TLX, as a result of higher levels of transparency [34]. They found in-depth SAT information brings a higher rating in the "physical workload" subscale of the NASA-TLX. The current study assessed how the transparency model interacts with agent reliability, and if that interaction will result in a concomitant change in workload.

### 2.3.5. Situation Awareness (SA)

Situation awareness (SA) refers to an individual's dynamic understanding of "what is going on" in a system [35]. In this model, SA is comprised of three hierarchical levels: perception of elements within the environment, the comprehension of their meaning, and a projection of their status in the near future [9,35]. In Level 3 automated vehicles, human operators need to maintain SA, and re-engage driving tasks if needed, so practical transparency information must be displayed to support the needs of awareness and control, while still maintaining the performance and cognitive benefits of automation [36]. As the SAT model was designed to support the operator's SA, the operator's SA was assessed to evaluate the effectiveness of the transparency model in the current study.

## 3. Current Study and Research Questions

Clearly, the relationship between agent transparency and reliability in driver-agent interaction is complex. Agent transparency could have a mitigating influence on the negative effects of agent reliability, and agent transparency's impact on human performance when automation is unreliable has yet to be explored. Little research has focused on both characteristics in one study to see whether there is the influence of transparency and reliability on drivers' performance outcomes. As such, the research questions we seek to examine are as follows.

- RQ 1. How will agent transparency influence drivers' perception of the agent, trust, workload, situation awareness, and takeover performance?
- RQ 2. How will agent reliability influence drivers' perception of the agent, trust, workload, situation awareness, and takeover performance?
- RQ 3. Are there any interaction effects of agent transparency and reliability on drivers' perception of the agent, trust, workload, situation awareness, and takeover performance?

To answer these research questions, the present study integrated transparency and reliability to systematically explore the effects of both critical features of in-vehicle intelligent agents on driver perception and performance in the context of conditionally automated driving. Multiple factors, particularly, drivers' perception of the agent, trust, perceived workload, situation awareness, and takeover performance have been investigated.

## 4. Methods

### 4.1. Participants

A total of 27 college students (nine females, 17 males, and one chose not to specify) with valid American driver's licenses participated in the study. Their age was between 19 and 30 (*Mean* = 21.46, *SD* = 2.42). Each of them was compensated with $10 for their time. Two participants' data were excluded from the study: one was excluded due to false driver's license information and the other was excluded due to breaking the rules during the experiment by maintaining manual driving all the time without required takeover or handover. The remaining 25 participants (eight female, 16 male, one declared as other, Mean age = 21.24, *SD* = 2.18) had an average driving experience of 4.32 years (*SD* = 1.57) and an average driving frequency of 5.88 times per week (*SD* = 4.32).

### 4.2. Apparatus

The study used a motion-based driving simulator (Nervtech[TM], Ljubljana, Slovenia). The simulator is equipped with three 48″ visual displays, a steering wheel, an adjustable seat, gas and brake pedals, and surrounded sound equipment. The driving scenarios were programmed in SCANeR studio, the software program that came with the simulator. Two humanoid robots, NAO (by SoftBank Robotics; Height: 22.6 in, Length: 12.2 in, Width: 10.8 in) and Milo (by Robokind; Height: 23.25 in, Length: 7.5 in, Width 9.75 in) were used as in-vehicle intelligent agents (See Figure 1). During driving, the robot was placed in the same fixed position to the participants' right (See Figure 2).

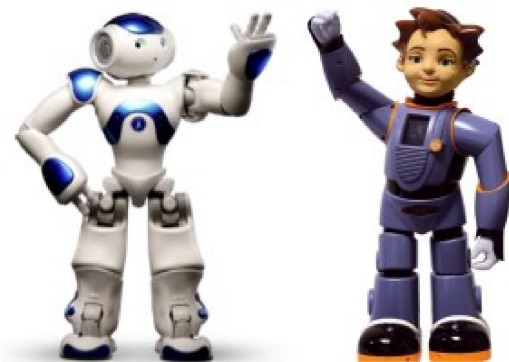

**Figure 1.** Robot NAO (**left**), robot Milo (**right**).

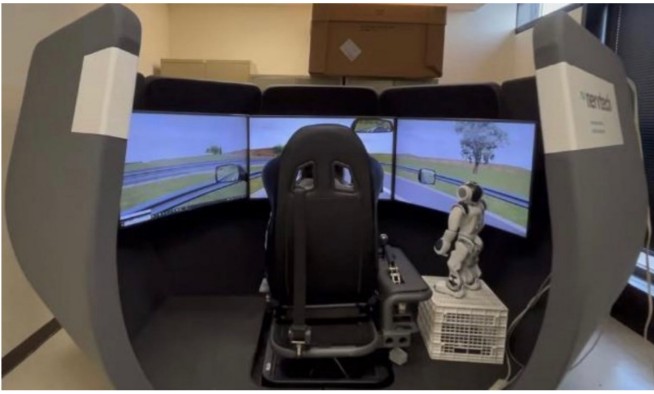

**Figure 2.** Experiment setup: Nervtech[TM] driving simulator with the IVIA.

### 4.3. Experimental Design

A 2 × 2 mixed factorial design was used in this study, with transparency (Push: proactive vs. Pull: on-demand) as a within-subjects variable and reliability (high vs. low) as a between-subjects variable. Thirteen participants were assigned to the high-reliability condition, where 100% of the information presented was accurate; while the remaining

twelve participants experienced the low-reliability condition, where 60% of the information presented was accurate. Tables 1 and 2 present the lists of agent intervention scripts in order. Within each reliability group, participants had two driving scenarios corresponding to two transparency conditions. They always experienced the Push transparency condition first, meaning that the agent provided all the information about the event in one string during the driving scenario they first experienced. The first statement uttered by the agent was a command or "what" statement, describing either what was occurring or what the participant was required to do (e.g., "Pedestrian on road ahead" or "Please take over"). Then, participants experienced the Pull transparency condition in the other scenario, where they were only provided with a single unit of information (e.g., "Please take over"), and the participants then had the opportunity to ask for more information up to two times by saying, "more information". (see Tables 1 and 2). The order of the transparent conditions was not counterbalanced deliberately, to show the potential information content availability under the Pull condition to the participants. Meanwhile, the learning effect was minimized by balancing the matching between the driving scenarios and the transparency conditions. In addition, each participant experienced both robots as their in-vehicle intelligent agents. The order in which each robot was assigned to the transparency condition was fully counterbalanced (See Table 3). All participants experienced the study in a driving simulator with both types of information transparency agents in turn (within-subjects condition): half of the participants had Milo as the Push-type agent and NAO as the Pull-type agent; the remaining half of the participants experienced the opposite. Further, the robots used for each reliability condition were counterbalanced. Half of the participants experienced a high reliable agent, and another half experienced a low reliable agent (between-subjects condition).

**Table 1.** Scenario 1 Intervention Scripts.

| Event No. | Description | Condition | |
| --- | --- | --- | --- |
| | | **Push** | **Pull** |
| | | **[Reliable/***Unreliable***]** [1] | **[Reliable/***Unreliable***]** |
| 1 | Construction Site | Please take over. The vehicle's front cameras detect an obstacle [**around a quarter mile**/*3 miles*] ahead | 1. Please take over<br>2. Obstacle around [**around a quarter mile**/*3 miles*] ahead<br>3. Detected by front cameras |
| 2 | Car Swerves | The car in front of you is expected to swerve into your lane in [**1000 feet**/*2 miles*] based on the system's prediction program. | 1. The car in front of you is expected to swerve into your lane<br>2. It is expected to swerve in [**1000 feet**/*2 miles*]<br>3. Detected by prediction program |
| 3 | Decision Error | Please take over. There is an error in the system's decision-making code.<br>. . . Never mind | 1. Please take over<br>2. System Error<br>3. Detected in system's decision code.<br>4. Never mind (given regardless of whether the participant asks for more info) |
| 4 | Jaywalker | The vehicle's front right sensors detect a [**pedestrian ahead who is walking into the street**/*large animal crossing the road ahead*]<br>Based on their trajectory the vehicle will brake and move to the left lane. | 1. [**Pedestrian on road**/*Animal crossing road*] ahead<br>2. Based on the pedestrian's trajectory, the vehicle will brake and move to the left lane<br>3. Detected by front right sensors |
| 5 | Fog | Please take over. The vehicle's light sensors detect heavy [**fog**/*rain*] ahead. | 1. Please take over<br>2. Heavy [**fog**/*rain*] ahead<br>3. Detected by vehicle's light sensors |

[1] Bolded part was presented in the high-reliability condition. The Italic part was presented in the low-reliability condition.

**Table 2.** Scenario 2 Intervention Scripts.

| | | Condition | |
|---|---|---|---|
| | | **Push** | **Pull** |
| **Event No.** | **Description** | [**Reliable**/*Unreliable*] [1] | [**Reliable**/*Unreliable*] |
| 1 | Construction Site | Please take over. The vehicle's front cameras detect an obstacle [**700 feet**/*2 miles*] ahead | 1. Please take over<br>2. Obstacle around [**700 feet**/*2 miles*] ahead<br>3. Detected by front cameras |
| 2 | Sensor Malfunction | The front right sensor is malfunctioning based on the test code.<br>. . . Never mind | 1. Please take over<br>2. Sensor malfunction<br>3. Detected in system's decision code.<br>4. Never mind (given regardless of whether the participant asks for more info) |
| 3 | Cow on road | The vehicle's front right sensors detect a [**large animal**/*child*] crossing the road ahead.<br>Based on their trajectory the vehicle will brake and move to the left lane. | 1. [**large animal**/*child*] on the road ahead<br>2. Based on the pedestrian's trajectory, the vehicle will brake and move to the left lane<br>3. Detected by front right sensors |
| 4 | Rain | Please take over. The vehicle's moisture sensors detect heavy [**rain**/*fog*] ahead. | 1. Please take over<br>2. Heavy [**rain**/*fog*] ahead<br>3. Detected by vehicle's moisture sensors |
| 5 | Car Swerves | The vehicle to your [**left**/*right*] is expected to swerve into your lane based on the system's decision model. | 1. Vehicle expected to move into your lane<br>2. The vehicle is positioned on your [**left**/*right*]<br>3. Detected in system's decision model |

[1] Bolded part was presented in the high-reliability condition. The Italic part was presented in the low-reliability condition.

**Table 3.** Experiment design matrix.

| Reliability | Transparency | Order of Robot (Scenario) | | | |
|---|---|---|---|---|---|
| | | **NAO (S1)** | **Milo (S2)** | **NAO (S2)** | **Milo (S1)** |
| High | Push | Group 1 | | Group 2 | |
| | Pull | | | | |
| Low | Push | Group 3 | | Group 4 | |
| | Pull | | | | |

S1 = Scenario 1, S2 = Scenario 2.

*4.4. Procedure*

Upon arrival, participants signed the consent form approved by Virginia Tech's Institutional Review Board (IRB) and completed a demographic questionnaire that included age, gender, and driving experience. Participants were informed that the vehicle is a conditionally automated vehicle and they are going to be accompanied by an intelligent agent during the driving scenario. Then, participants underwent a simulation sickness test following the Georgia Tech Simulator Sickness Screen Protocol [37], including 3 min of test drive and self-comfort checklists before and after the test drive. Before the test drive, they were told that the scene would be randomly paused during the formal drive, and they would be asked to answer questions. They had a practice of the pause during the test drive. This process allowed them to get familiar with the driving simulator, making sure that they do not have simulator sickness and were safe for the study. The participants who only did not get simulator sickness continued to conduct an actual experiment. No participants reported that they had simulator sickness during the test drive. Then, participants were presented with a thorough explanation of the experimental procedures before the beginning of each trial. After completing each trial, participants filled out the NASA-TLX scale and subjective

questionnaires. After finishing two trials, their preference toward the two transparency conditions was collected.

### 4.5. Dependent Measures

Both subjective measures and objective measures were used in the study. Subjective measures included rating scales used to assess the interaction between driver and agent, which are (1) Godspeed questionnaire, containing five factors with 24 items on a 5-point semantic differential scale [26]; (2) Social presence questionnaire: five items on a 10-point Likert scale [38]; (3) Robotic Social Aptitude Scale Questions (RoSAS): three factors with 18 items on a 7-point Likert scale [39]; (4) Subjective Assessment of Speech System Interfaces (SASSI): six factors with 34 items on a 7-point Likert scale [40]; (5) Scale of Trust in Automated Systems: twelve items on a 7-point Likert scale [41]. Participants' perceived workload was collected through NASA-TLX [33]. In addition, participants' preferences toward two types of transparency and their reasons were collected through a questionnaire.

Objective measures were obtained from Situation Awareness Global Assessment Technique (SAGAT) and the driving simulator.

SAGAT queries were used to capture drivers' situation awareness (SA) of their environment [35]. It is a freeze-probe recall method that involves asking participants of their awareness of the SA elements during random freezes of the simulated environment. Research shows that SAGAT does not distract the participant from the main task of driving [42].

In both scenarios, a simulation scenario was paused at a pre-defined place without participants' awareness beforehand about when, where, and how many times it would happen; the screen was hidden from the driver immediately following the pause. During each pause, participants were asked to answer a paper questionnaire that was designed based on the design guideline of SAGAT [22], to measure all three situation awareness levels (perception, comprehension, and projection) of the driver. The questionnaire contained six questions in total, including two questions about Level 1 SA, two questions about Level 2 SA, and two questions about Level 3 SA. After all the answers were scored, the average score for each SA level was calculated to analyze situation awareness for each level. The average score of SA was calculated from an average score of the three SA levels.

The driving simulator produced raw CSV data, containing takeover time, speed, acceleration, and steering wheel angles [33].

Takeover time: The response time of takeover was measured from the time the agent starts the command and the first input of the driver takes over control (either braking or turning the wrench near the steering wheel), measured in seconds.

Takeover quality [32]:

1. Speed: the maximum/minimum/average speed during the takeover, measured in kilometers per hour.
2. Acceleration: the maximum/minimum/average longitudinal and lateral acceleration during the takeover, measured in $m/s^2$.
3. Steering Wheel Angle: the maximum/standard deviation of steering wheel angle of the takeover period, measured in degrees.

### 4.6. Data Analysis

Repeated-measures analysis of variance (ANOVA) was conducted to investigate the difference between participants' NASA-TLX workload, situation awareness, subjective survey data, and driving performance. The driving performance includes takeover performance (takeover time, maximum/minimum/average/standard deviation of speed, acceleration, and wheel angle during participants take over the control), the number of requests for more information in the Pull-type transparency condition (Pull request), and the number of compliances of takeover requests. All statistical analyses were performed by JMP Pro 16.

## 5. Results

### 5.1. User Perception and Trust in Automation

Results from $2 \times 2$ repeated measures ANOVA indicated that transparency and reliability influenced user perception in different ways. Tables 4 and 5 show the statistical analysis of the response.

**Table 4.** Subjective evaluation results: condition means and standard deviation.

| Scale | Items | Agent Condition Mean (SD) | | | |
|---|---|---|---|---|---|
| | | H [1], Push | H, Pull | L [1], Push | L, Pull |
| GodSpeed | Anthropomorphism | 2.33 (0.77) | 2.93 (1.04) [†] | 2.73 (0.72) | 2.65 (0.87) |
| | Animacy | 2.60 (0.85) | 3.22 (1.05) [†] | 2.69 (0.89) | 2.86 (0.91) |
| | Likeability | 3.62 (1.01) | 4.05 (0.91) [†] | 3.51 (0.74) | 3.63 (1.02) |
| | **Perceived Intelligence *** | 4.17 (0.73) [†] | 3.89 (1.03) | 3.92 (0.67) | 3.74 (0.81) |
| | Perceived Safety | 3.64 (0.52) | 3.63 (0.83) | 3.59 (0.64) | 3.51 (0.59) |
| Social Presence | Social Presence | 4.90 (1.44) | 5.65 (1.36) | 5.61 (1.15) | 5.57 (1.04) |
| RoSAS | **Competence *** | 5.54 (1.36) [†] | 5.25 (1.61) | 5.08 (0.83) | 4.64 (1.38) |
| | **Warmth *** | 2.64 (1.75) | 4.18 (1.79) [†] | 3.05 (1.16) | 3.17 (1.55) |
| | Discomfort | 2.17 (0.98) | 2.21 (1.66) | 2.19 (0.74) | 2.13 (1.67) |
| SASSI | System Response Accuracy | 5.03 (0.83) | 5.30 (1.34) | 4.90 (0.84) | 4.71 (1.13) |
| | Likeability | 4.96 (1.16) | 5.29 (1.40) [†] | 5.19 (0.87) | 4.91 (1.38) |
| | Cognitive Demand | 3.20 (1.37) | 3.17 (1.23) | 3.20 (1.09) | 3.09 (1.20) |
| | Annoyance | 3.00 (1.80) | 2.85 (1.52) | 2.85 (1.29) | 2.95 (1.47) |
| | Habitability | 3.96 (0.58) | 4.34 (1.22) [†] | 3.94 (0.64) | 4.13 (0.55) |
| | Speed | 4.75 (0.97) [†] | 4.17 (2.33) | 4.69 (1.38) | 4.08 (1.98) |
| Trust in Automation | Trust | 4.54 (1.26) | 5.23 (1.37) [†] | 4.64 (1.12) | 4.73 (0.93) |

[1] H = High Reliability, L = Low Reliability, [†] Top score, * $p < 0.05$.

**Table 5.** Subjective evaluation statistics.

| Scale | Items | F Value and Significance ($p$) | | | | | |
|---|---|---|---|---|---|---|---|
| | | Transparency | | Reliability | | Transparency $\times$ Reliability | |
| | | F | p | F | p | F | p |
| GodSpeed | Anthropomorphism | 1.75 | 0.20 | 0.15 | 0.70 | 3.73 | 0.07 |
| | Animacy | 3.40 | 0.08 | 0.15 | 0.70 | 1.32 | 0.26 |
| | Likeability | 1.78 | 0.19 | 0.77 | 0.39 | 0.54 | 0.47 |
| | **Perceived Intelligence** | 4.32 | **0.049 *** | 0.91 | 0.35 | 0.86 | 0.36 |
| | Perceived Safety | 0.10 | 0.76 | 0.14 | 0.70 | 0.09 | 0.76 |
| Social Presence | Social Presence | 1.02 | 0.32 | 1.32 | 0.26 | 2.52 | 0.13 |
| RoSAS | **Competence** | 4.81 | **0.039 *** | 0.96 | 0.34 | 0.38 | 0.54 |
| | **Warmth** | 4.53 | **0.043 *** | 0.49 | 0.49 | 3.07 | 0.09 |
| | Discomfort | 0.00 | 0.98 | 0.01 | 0.92 | 0.02 | 0.88 |
| SASSI | System Response Accuracy | 0.04 | 0.84 | 0.99 | 0.33 | 1.46 | 0.24 |
| | Likeability | 0.00 | 0.95 | 0.04 | 0.84 | 1.20 | 0.28 |
| | Cognitive Demand | 0.07 | 0.79 | 0.00 | 0.95 | 0.02 | 0.89 |
| | Annoyance | 0.00 | 1.00 | 0.04 | 0.84 | 0.22 | 0.64 |
| | Habitability | 1.52 | 0.23 | 0.01 | 0.90 | 1.05 | 0.32 |
| | Speed | 2.34 | 0.14 | 0.06 | 0.81 | 0.01 | 0.90 |
| Trust in Automation | Trust | 2.50 | 0.13 | 0.39 | 0.53 | 1.46 | 0.24 |

* $p < 0.05$.

Transparency had a significant main effect on (1) Perceived Intelligence: $F(1, 24) = 4.32$, $p = 0.049$, $\eta_p^2 = 0.19$, which is the subscale of Godspeed. The Push-type agents were perceived more intelligent (*Mean* = 4.04, *SD* = 0.70) than the Pull-type agents (*Mean* = 3.80, *SD* = 0.90). (2) Competence: $F(1, 24) = 4.81$, $p = 0.039$, $\eta_p^2 = 0.15$, which is the subscale of RoSAS. The Push-type agents were perceived more competent (*Mean* = 5.3, *SD* = 1.12) than the Pull-type agents (*Mean* = 4.93, *SD* = 1.50). (3) Warmth: $F(1, 24) = 4.53$, $p = 0.044$, $\eta_p^2 = 0.17$, which is the subscale of SASSI. The Pull-type agents were perceived warmth, presenting more emotion (*Mean* = 3.65, *SD* = 1.71) than the Push-type agents (*Mean* = 2.86, *SD* = 1.46). No significant difference between transparency conditions was found in other scales (see Table 5).

No significant main effect of reliability and interaction effect between transparency and reliability was found on other items of the questionnaires (see Table 5).

### 5.2. Takeover Performance

The result from repeated measures ANOVA indicated an interaction effect between transparency and reliability on maximum lateral acceleration: $F(1, 24) = 5.34$, $p = 0.03$, $\eta_p^2 = 0.063$ (see Figure 3). With high reliability, the Push-type agent (*Mean* = 1.11, *SD* = 0.82) led to significantly higher maximum lateral acceleration than the Pull-type agent (*Mean* = 0.75, *SD* = 0.61) $t(30) = 2.067$, $p < 0.05$. With low reliability, the Pull-type agent (Mean = 1.11, SD = 1.02) led to numerically higher maximum lateral acceleration than the Push-type agent (*Mean* = 0.96, *SD* = 0.65) $t(35) = -0.856$, $p > 0.05$, but this did not lead to the statistically significant difference. There were no significant results on takeover time, maximum/minimum/average speed or steering wheel angles (see Table 6).

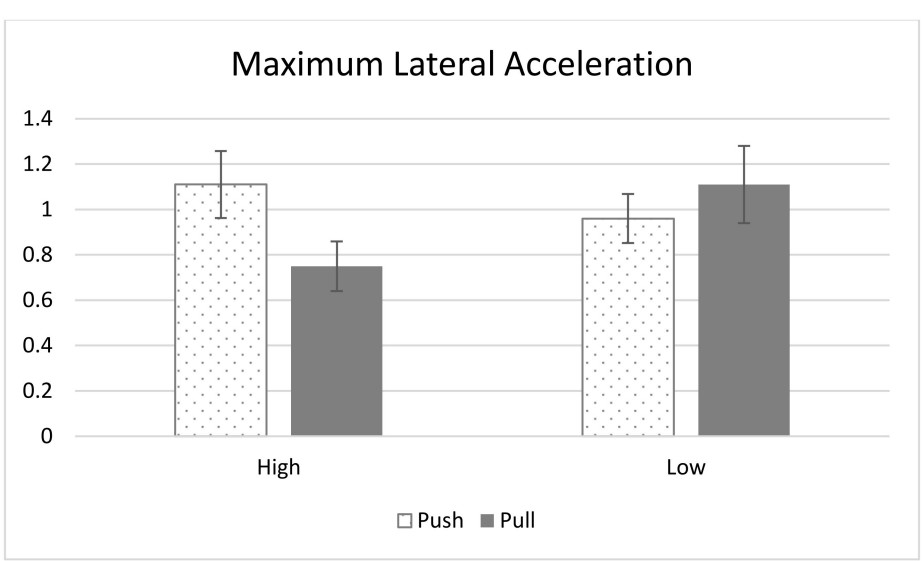

**Figure 3.** Maximum Lateral Acceleration ($p < 0.05$) [error bars indicate standard errors].

**Table 6.** Takeover performance statistics.

| Takeover Measures | Items | F Value and Significance (*p*) | | | | | |
|---|---|---|---|---|---|---|---|
| | | Transparency | | Reliability | | Transparency × Reliability | |
| | | *F* | *p* | *F* | *p* | *F* | *p* |
| Take over time | Take over time | 2.38 | 0.14 | 0.01 | 0.91 | 0.26 | 0.62 |
| Speed | Maximum | 1.01 | 0.33 | 0.47 | 0.50 | 0.91 | 0.33 |
| | Minimum | 0.07 | 0.80 | 1.78 | 0.20 | 0.03 | 0.86 |
| | Average | 0.63 | 0.44 | 1.04 | 0.32 | 0.30 | 0.59 |

**Table 6.** *Cont.*

| Takeover Measures | Items | F Value and Significance (*p*) | | | | | |
|---|---|---|---|---|---|---|---|
| | | Transparency | | Reliability | | Transparency × Reliability | |
| | | *F* | *p* | *F* | *p* | *F* | *p* |
| Longitudinal Acceleration | Maximum | 0.02 | 0.88 | 0.80 | 0.38 | 1.43 | 0.24 |
| | Minimum | 0.02 | 0.90 | 0.02 | 0.90 | 1.04 | 0.32 |
| | Average | 0.08 | 0.77 | 0.69 | 0.41 | 1.71 | 0.20 |
| Lateral Acceleration | **Maximum** | 0.42 | 0.52 | 0.15 | 0.70 | 5.34 | **0.03 \*** |
| | Minimum | 4.05 | 0.06 | 0.00 | 1.00 | 1.78 | 0.19 |
| | Average | 1.01 | 0.33 | 0.17 | 0.68 | 1.69 | 0.21 |
| Wheel Angel | Maximum | 0.60 | 0.45 | 0.31 | 0.59 | 0.32 | 0.60 |
| | Standard deviation | 0.51 | 0.48 | 0.02 | 0.89 | 0.02 | 0.89 |

\* $p < 0.05$.

### 5.3. Agent Preference

Figure 4 shows the participants' preference ranking for agents between two types of transparency conditions. Compared with the Pull condition, the Push condition was ranked as their first choice, with 84 percent of people choosing it.

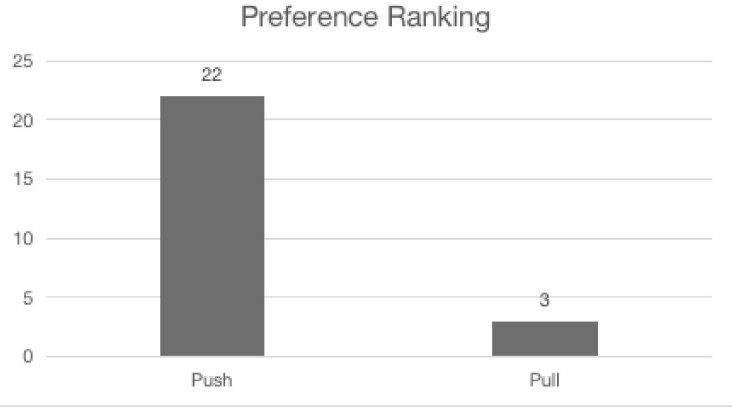

**Figure 4.** Preference ranking of agent's transparency.

Participants provided their reasons to explain their preference for Push-type agents:

"Providing the information automatically with the 'please take over' was much more useful as I didn't have to process much beyond that, like thinking about what I would say." (P2, chose Push)

"I want to confirm if the system detects the situation correctly." (P3, chose Push)

"When I ask for more information, it might take a little too much time. It is easier to know the reason for caution earlier on and is less stressful in that way." (P23, chose Push)

However, not all participants liked the Push-type agents. For the three participants who chose the Pull-type agent, two of them provided explanations that they were able to see what is ahead and ready to take over anytime without further details from the agent.

### 5.4. Pull Request

There was no significant difference in requesting more information between high reliability (*Mean* = 1.92, *SD* = 1.62) and low reliability conditions (*Mean* = 1.85, *SD* = 2.97), $F(1, 24) = 0.0053$, $p = 0.94$ based on repeated measures ANOVA.

It is important to note that the Pull request in each trial can be up to 10 times. Numerically, the results show that in the Pull condition, the request time for more information in both reliability conditions was less than 20%.

### 5.5. Workload

Results from repeated measures of ANOVA indicated a significant main effect of transparency on physical demand, $F(1, 24) = 6.42$, $p = 0.02$, $\eta_p^2 = 0.22$ (see Figure 5, Table 7). Under the Pull condition, participants showed more physical demand to accomplish the task (*Mean* = 23, *SD* = 14.36), compared with participants under the Push condition (*Mean* = 18.8, *SD* = 11.84).

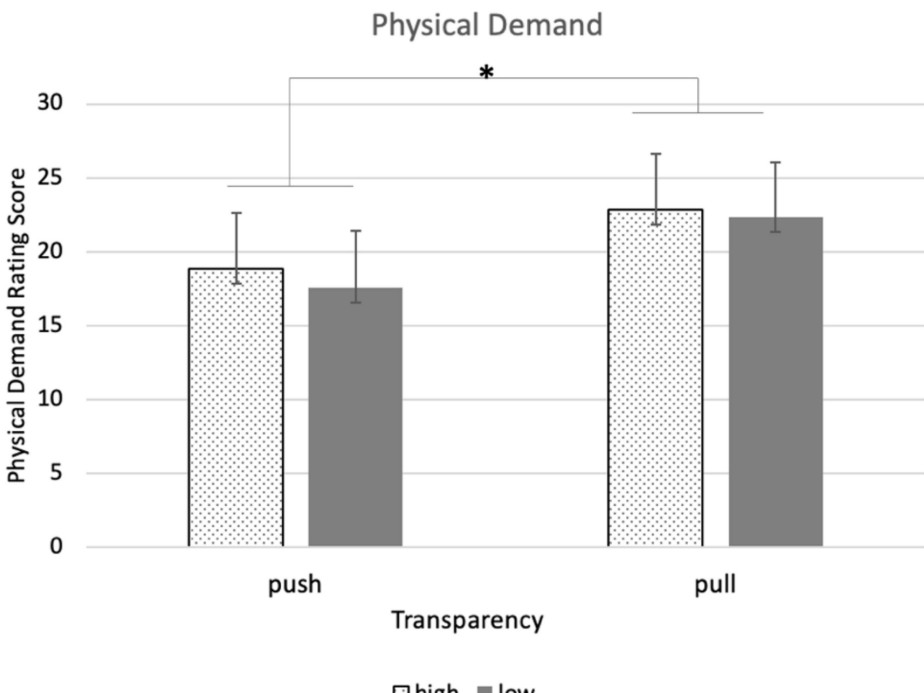

**Figure 5.** Physical demand measured by NASA/TLX in all conditions (* $p < 0.05$) [error bars indicate standard errors].

**Table 7.** NASA/TLX statistics.

| Items | F Value and Significance (*p*) | | | | | |
| --- | --- | --- | --- | --- | --- | --- |
| | Transparency | | Reliability | | Transparency × Reliability | |
| | *F* | *p* | *F* | *p* | *F* | *p* |
| Mental Demand | 0.26 | 0.61 | 0.64 | 0.43 | 0.09 | 0.77 |
| **Physical Demand** | 6.42 | **0.02 *** | 0.03 | 0.86 | 0.04 | 0.86 |
| Temporal Demand | 0.24 | 0.62 | 0.01 | 0.93 | 0.01 | 0.95 |
| Performance | 0.16 | 0.70 | 0.02 | 0.89 | 1.74 | 0.20 |
| **Effort** | 0.42 | 0.52 | 5.57 | **0.03 *** | 0.17 | 0.68 |
| Frustration | 2.52 | 0.13 | 0.07 | 0.79 | 0.21 | 0.65 |
| Total Workload | 0.94 | 0.34 | 0.72 | 0.40 | 0.49 | 0.49 |

* $p < 0.05$.

Results from repeated measures ANOVA indicated a significant main effect of reliability on effort, $F(1, 24) = 5.57$, $p = 0.03$, $\eta_p^2 = 0.17$ (see Figure 6). Under the low-reliability condition, participants gave more effort to accomplish their level of performance (*Mean* = 41.00, *SD* = 23.27), compared with participants under the high-reliability condition (*Mean* = 26.4, *SD* = 15.11).

No interaction effect between transparency and reliability was found in any of the subscales of the NASA-TLX (see Figure 7).

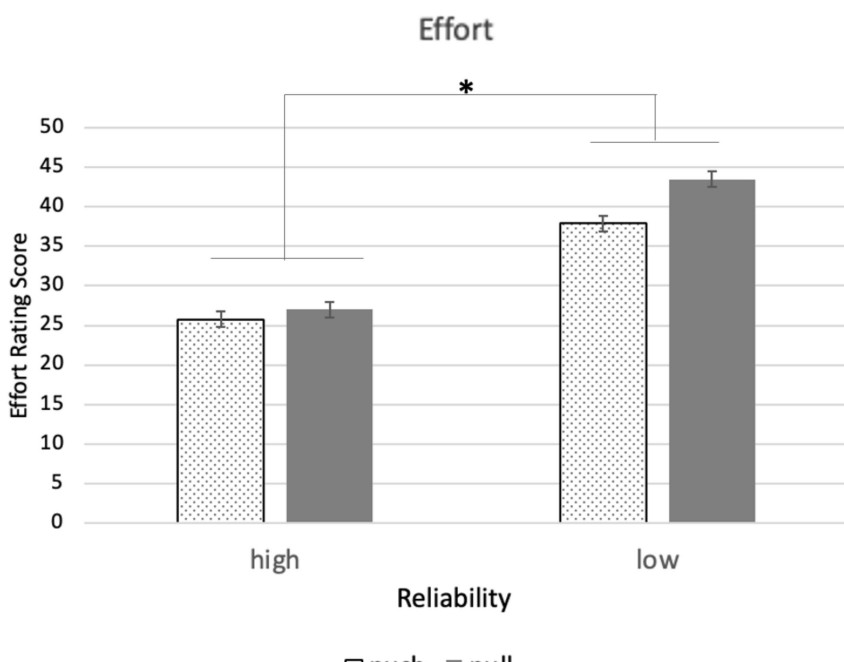

**Figure 6.** Effort measured by NASA/TLX in all conditions (* *p* < 0.05) [error bars indicate standard errors].

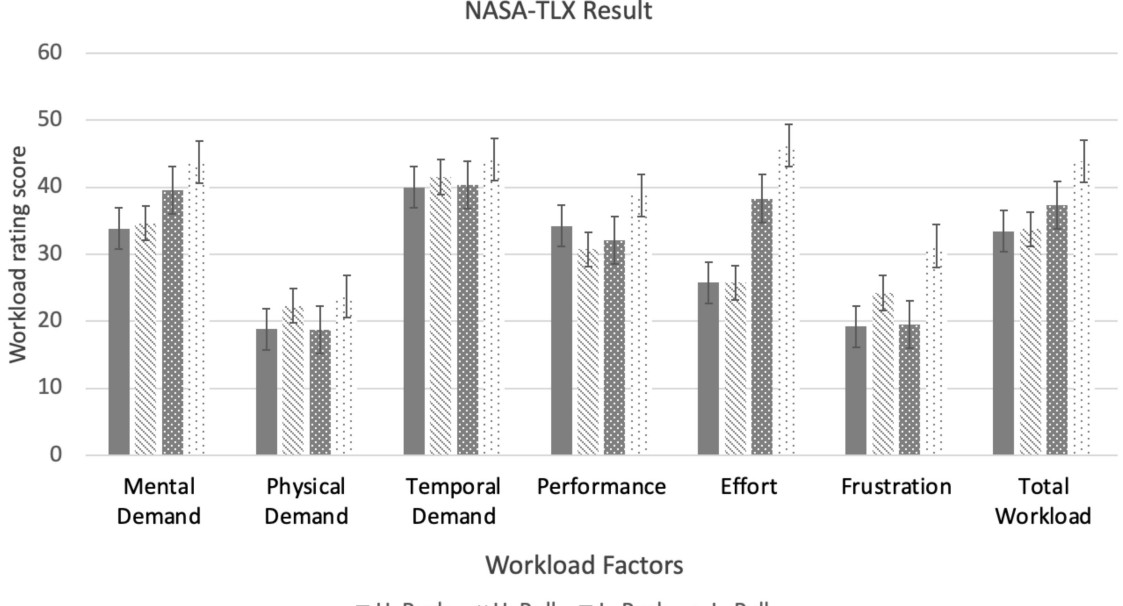

**Figure 7.** NASA/TLX Result [error bars indicate standard errors].

### 5.6. Situation Awareness (SAGAT)

The result from repeated measures of ANOVA indicated a significant main effect of reliability on Level 1 SA, $F(1, 24) = 4.93$, $p = 0.036$, $\eta_p^2 = 0.146$. Under the high-reliability condition, participants had a higher perception of the elements in the environment (*Mean* = 0.81, *SD* = 0.20) compared to participants under the low-reliability condition (*Mean* = 0.64, *SD* = 0.33).

There was no significant difference in average rating scores in their Level 2 SA or Level 3 SA (Table 8). However, under the high-reliability condition, participants had numerically higher Level 2 and Level 3 SA scores compared to participants under the low-reliability condition (see Table 9).

**Table 8.** Situation Awareness statistics.

| Items | F Value and Significance ($p$) | | | | | |
|---|---|---|---|---|---|---|
| | Transparency | | Reliability | | Transparency $\times$ Reliability | |
| | $F$ | $p$ | $F$ | $p$ | $F$ | $p$ |
| **Level 1 SA** | 2.85 | 0.10 | 4.93 | **0.036 *** | 0.00 | 0.95 |
| Level 2 SA | 0.50 | 0.49 | 0.67 | 0.42 | 0.00 | 0.97 |
| Level 3 SA | 0.76 | 0.39 | 1.84 | 0.19 | 0.36 | 0.55 |
| Average | 2.36 | 0.14 | 2.50 | 0.13 | 0.06 | 0.81 |

* $p < 0.05$.

**Table 9.** SAGAT result (mean and standard deviation).

| Situation Awareness | Condition Mean (SD) | |
|---|---|---|
| | High Reliability | Low Reliability |
| **Level 1 SA *** | 0.81 (0.20) | 0.64 (0.33) |
| Level 2 SA | 0.56 (0.44) | 0.46 (0.37) |
| Level 3 SA | 0.45 (0.35) | 0.30 (0.30) |
| Average | 0.60 (0.27) | 0.47 (0.26) |

* $p < 0.05$.

*5.7. Compliance*

Table 10 shows the statistical analysis of the compliance of the takeover request. There was no significant main effect in compliance on either transparency, $F(1, 25) = 0.03$, $p = 0.84$, $\eta_p^2 = 0.004$, or reliability, $F(1, 24) = 0.47$, $p = 0.50$, $\eta_p^2 = 0.026$, based on repeated measures ANOVA. There was no interaction effect in compliance between reliability and transparency, $F(1, 24) = 3.39$, $p = 0.08$, $\eta_p^2 = 0.072$.

**Table 10.** Mean and standard deviation of the number of compliances in each condition.

| | Transparency | |
|---|---|---|
| Reliability | High Reliability, Push 2.54 (0.18) | High Reliability, Pull 2.92 (0.19) |
| | Low Reliability, Push 3.00 (0.19) † | Low Reliability, Pull 2.70 (0.25) |

† Top score.

It is important to note that the maximum compliance of takeover requests in each trial is three times. Numerically, the result shows that compliance time is rather high in all cases, with the lowest being the Push condition with high-reliability information (84%) and the highest being the Push condition with low-reliability information (100%).

## 6. Discussion

The objective of the study was to investigate the effects of in-vehicle intelligent agents' transparency and reliability on drivers' perception, takeover performance, workload, and situation awareness in conditionally automated vehicles. Results showed that the agent's transparency seemed to contribute to participants' perception towards the automated vehicle system, perceived workload, and takeover performance. On the other hand, the agent's reliability had a significant influence on driver perceived workload and SA. There was a significant interaction effect on takeover performance.

- RQ 1. How will agent transparency influence drivers' perception of the agent, trust, workload, situation awareness, and takeover performance?

To answer RQ1, the results showed that transparency significantly influenced drivers' perception of the agent, including perceived intelligence, competence, and warmth. Further, participants showed a higher preference on the Push-type agent than the Pull-type agent.

Results showed that participants perceived the Push-type agent more intelligent and competent than the Pull-type agent; whereas they felt significantly more warmth from the Pull-type agent than the Push-type agent, regardless of the reliability level. Prior research argued that participants consider an agent more intelligent when it provides more information [10], which was consistent with our result that participants found the Push-type agent more intelligent. The warmth that Pull-type agent brought to the participants can be explained by prior research suggesting that a chatbot which develops more conversation promotes perceived humanness [43]. Note that the Pull agent required the participants to communicate with the agent more. In the same line, a numerical trend appears in other subjective questionnaires that under the high-reliability condition, the Push-type agent received the highest rating for perceived intelligence, competence, and system response speed, which are associated with the *performance*-based aspect of the system; whereas the Pull-type agent received the highest rating for anthropomorphism, animacy, likeability, warmth, habitability, and trust, which are more associated with the *relationship*-based aspect of the system. Because driving is a safety-critical task situation, the participants might have chosen the performance-first agent, which can lead to better safety, rather than the social and relational agents. This might explain why more participants chose the Push-type agent as their preference. Among the reasons that participants addressed for choosing the "Push" agent, most expressed a desire to confirm the accuracy of the system.

On the other hand, results showed that transparency had a significant influence on the participants' physical demand of workload. Previous research has shown that increased information content does not necessarily lead to higher workload when working with a reliable agent [15,44]. Based on that, the difference in physical demand of the present study might not come from the different amount of information in the two conditions. However, the difference might come from the fact that the Pull-type agent asked the participants for more interactions. Thus, designing the interaction mode might be more important than deciding the amount of information for drivers' perceived workload.

These outcomes may be beneficial for designers and developers when designing transparency and assigning proper transparency considering usage scenarios when it comes to the interaction between drivers and in-vehicle intelligent agents.

- RQ 2. How will agent reliability influence drivers' perception of the agent, trust, workload, situation awareness, and takeover performance?

Compared to agents with low reliability, agents with high reliability showed significantly less effort for participants to accomplish their performance and higher situation awareness. This supports the previous finding [45] that human operators' performance improves with the automation reliability in general.

Surprisingly, the result shows that the reliability does not influence the result of the questionnaire, such as accuracy (SASSI) and trust (Trust in automation) or behavioral compliance with the system. One possible reason could be the potentially high initial trust in the automated system. Prior research has shown that users may begin their work with an unfamiliar system with overly high expectations of the system's performance (i.e., positive bias [46,47]). Thus, considering that most of our participants were novices (22/25) to the simulator, although we did not measure initial trust in the study, participants might have a relatively high level of initial trust in the automated vehicle they drove. This could explain why they had a higher workload on effort when they received low-reliability information; they might feel confused and need to make more effort to respond [48]. In the low reliability condition, due to the conflict between the environmental information perceived by the driver and the information given by the agent, the driver may have needed more cognitive resources than in the high reliability condition. According to the concept of limited attentional resource [49], the driver's workload increased and situation awareness might be damaged accordingly. Even though the high reliability agent enhanced driver situation awareness only at Level 1, the same trend appeared at higher levels of SA, showing that the information in the high reliability condition helped our participants gain better situation awareness, overall. However, this different SA scores did not lead

to significant performance differences. Note that good SA does not guarantee good or better performance because in between there are at least two more steps–decision-making and action selection. Further, the higher initial trust could help to explain why reliability did not influence the rating of the trust scale. This explanation was also confirmed in the analysis of compliance, with a high average number of compliance responses across the different reliability conditions.

The other possible explanation is that high transparency might diminish the negative effect of unreliability on trust. In the two scenarios we designed, the system informed about the system failure of certain events (i.e., "decision error" in Scenario 1, and "sensor malfunction" in Scenario 2) which might serve as a safeguard for temporary agent trust reduction when experiencing an actual malfunction [50]. By receiving sufficient information from XAI, users gained the chance to better anticipate system behavior and would be able to adapt their expectations early in the process and thus, might not be negatively surprised when the malfunction occurred. This finding is also in line with earlier studies showing that appropriate information about system functioning may lead to a facilitated trust calibration when system malfunctions occur [51,52].

- RQ 3. Are there any interaction effects of agent transparency and reliability on drivers' perception of the agent, trust, workload, situation awareness, and takeover performance?

There was only one significant interaction effect on the maximum lateral acceleration of takeover performance. A combination of high reliability and high transparency level led to higher maximum lateral acceleration, and a combination of low reliability and low transparency level also led to the similar outcome. We can cautiously infer that providing more information when the system is highly reliable might increase the participants' confidence; while providing less information when the system is unreliable might increase the participants' urgency.

Other than that, no significant interaction effect of agent transparency and reliability was found on the participants' perception of the agent, trust, perceived workload, or situation awareness.

## 7. Limitations and Future Work

Future research on this topic can disentangle the underlying mechanisms by considering the limitations of the current study.

First, the order of the transparency conditions was not counterbalanced, and the order of the Push and Pull conditions might influence the results. The order of transparency was designed in such a way (i.e., Push-type first and Pull-type second) that participants could learn from the Push condition about what kind of information is available in the Pull condition thereafter.

Second, in both transparency conditions, the actual information amount was the same if the participants requested more information. In our experiment, we still succeeded in manipulating the transparency of the system because the number of the request for more information in both reliability conditions was less than 20%. However, different approaches to the transparency manipulation will add interesting insights (e.g., differences in a total amount of information provided).

Third, our study lies in the use of the driving simulator. The driving simulator provides a reliable method for manipulating the experiment and a safe environment for testing in the lab. Although the simulator we used in the current study is a motion-based one with the surrounding sound equipment that could mimic a real driving experience to a large extent, there is still some amount of difference in fidelity between the simulation and real-life driving. Conducting driving research in real traffic and road conditions would increase the reliability and validity of the data regarding both attitudes and performance.

Another limitation of our study is that our participant number may not be sufficient to show all the possible outcomes, and we are planning to replicate and extend the current study with more participants in the future.

The potential interaction effect between transparency and reliability is worthy of further measurement and investigation in the future. Further investigation should treat some of the potential factors mentioned in the paper that may affect the result (e.g., information about the system's weakness may have an influence on initial trust), helping to achieve further understanding of the effects of transparency and reliability on drivers' outcomes in automated vehicles. Further, this experiment could be expanded to include dynamic trust measures by adding measures for a pre- and post-trust investigation to test the initial trust hypothesis discussed in the current study. Physiological sensors can be added to obtain an objective measurement of workload and trust, and their dynamic calibration with the agent's transparency and reliability. Further, this study found a trend of high subjective rating scores on performance-related items in the Push conditions, and social relationship-related items in the Pull conditions when the agent was highly reliable. In this line, questionnaires such as cognitive trust and affective trust can be used to assess the scientific consensus and practical merits of the two variables.

**Author Contributions:** Conceptualization, M.J.; methodology, M.J. and J.Z.; software, J.Z.; formal analysis, J.Z.; resources, M.J.; data curation, J.Z.; writing—original draft preparation, J.Z.; writing—review and editing, J.Z. and M.J.; supervision, M.J.; project administration, M.J. All authors have read and agreed to the published version of the manuscript.

**Funding:** This research received no external funding.

**Institutional Review Board Statement:** The study was conducted in accordance with the Declaration of Helsinki, and approved by the Institutional Review Board of Virginia Tech (protocol code 19-088/VT19-088-568(TRX), approved on 02/09/2022).

**Informed Consent Statement:** Informed consent was obtained from all subjects involved in the study.

**Data Availability Statement:** Not applicable.

**Conflicts of Interest:** The authors declare no conflict of interest.

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
