# Peer review of "The Effects of Transparency and Reliability of In-Vehicle Intelligent Agents on Driver Perception, Takeover Performance, Workload and Situation Awareness in Conditionally Automated Vehicles"

_mti, doi:10.3390/mti6090082_

Round 1
Reviewer 1 Report
The authors present a study about the use of in-vehicle intelligent agents in the context of automated vehicles. Specifically, they evaluated the impact of two interaction strategies, that is, push VS pull, on driver's perception of the agent and the takeover time with respect to the self-driving car.
They use a rigorous setup and a well-defined methodology to conclude that pull leads to a better outcome.
The paper is very well written and scientifically sound.
On page 8, section 4.5, line 269 there is a missing reference.
Section 6 is very long and has a lot of content. I would recommend moving the limitations and future work to a separate "Conclusions" section.
Reviewer 2 Report
Actually, I am not familiar with the in-vehicle intelligent agents (IVIAs). Nevertheless, the research topic sounds very interesting and also challenging. The problem definition is clear. The structure of the paper is easy to follow. There are just few comments for some minor changes and future study.
1. The authors need to highlight the contribution of the paper in the last paragraph of 1. Introduction in the revised manuscript. In addition, it seems that the organization of the paper is necessary to be added in the introduction section.
2. There is a hyperlink error ("Click or tap here to enter text") in the second paragraph of page 8. Please revise the manuscript.
3. I would like to suggest the authors to gather more participants to conduct statistical analyses in the future study, which may provide a strong evidence to support the results of the current study.
